# How Company Characteristics Influence Measurement Practices and Disclosure Level Prescribed within IAS 41

Mohammad Saleh Altarawneh 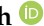

Accounting Department, Mutah University, Karak 61710, Jordan; m.s.tarawneh@mutah.edu.jo

**Abstract:** This research paper describes the accounting practices of Jordanian companies engaged in agricultural activities, and identifies the influence of company characteristics on measurement practices related to asset pricing and level of disclosure required by IAS 41. Company characteristics were considered as: size, intensity of biological assets (BA), level of international activities, and audit for the Big Four. Dependent variables were considered measurement practices related to valuing BA as well as resultant harvest and disclosure level, the latter being measured by mandatory and voluntary disclosures. The entire population of companies that include one or more agricultural activities in their purposes and are considered reporting companies formed the research sample, giving a total of 259 companies. The findings revealed that both intensity of BA and level of international activities have a positive impact on measurement practices. Audit for the Big Four was the strongest variable influence, the overall disclosure level prescribed by IAS 41, followed by the level of international activities variable. However, the intensity of the BA variable affects only the overall disclosure level for companies that measure their BA based on the cost method. Firm size was found to have no influence on either measurement practices or disclosure level. The key value of this paper is its examination of the role of company characteristics on measurement practices and level of disclosure required by IAS 41 in the context of Jordanian companies. Through this examination, this study is helpful to standards setters and regulators who obligate and issue the financial regulation and reporting standards at a national or international level, supporting their understanding of measurement and disclosure practices adopted in agricultural companies in the developing country context of Jordan.

**Keywords:** IAS 41; biological assets (BA); cost model; fair value (FV); company characteristics; measurement; disclosure; Jordan

## 1. Introduction

Despite agricultural commodities representing only a fraction of most countries' gross domestic product (GDP), problems pertaining to agriculture significantly influence economies and their respective societies. As a primary industry, agriculture produces grains, fibers and biological assets, deemed as key inputs to several industries, for example those involving intensive agricultural projects whether confined or not, food and clothes production, the cosmetics sector and pharmaceuticals. These industries transform raw materials to produce finished goods that assist in building wealth for either shareholders in particular or economies in general. However, it is evident that natural resources may be depleted with the result that companies might encounter significant risk and loss (ACCA et al. 2012; Dasgupta 2008; Costanza et al. 2014).

Consequently, it is essential to properly manage the depletion and consumption of these resources by taking global actions to protect the wealth of humanity. At the company level, requirements must be followed related to the environment and reports of how these are satisfied must be included as part of their disclosures on corporate social responsibility. Such practices increase the value relevance of information and, as noted by Ogilvy (2015),

the measurement of biological assets (bearer plants, livestock, the harvest and produce) is required by all accounting bodies.

In most countries, agricultural enterprises are primarily regulated under International Accounting Standards (IAS) number 41 Agriculture, as "*Agricultural activity is the management by an entity of the biological transformation and harvest of biological assets for sale or for conversion into agricultural produce or into additional biological assets*" (IFRS 2021a, IAS 41, para. 5).

Biological assets (BA) include livestock, crops, and fruits. Changes in the physical characteristics of living animals or plants directly increase or decrease the economic benefits to the business (Deloitte 2000). "*This Standard is applied to agricultural produce, which is the harvested produce of the entity's biological assets, at the point of harvest. Thereafter, IAS 2 Inventories or another applicable Standard is applied. Accordingly, this Standard does not deal with the processing of agricultural produce after harvest; for example, the processing of grapes into wine by a vintner who has grown the grapes. While such processing may be a logical and natural extension of agricultural activity, and the events taking place may bear some similarity to biological transformation, such processing is not included within the definition of agricultural activity in this Standard*" (IFRS 2021a, IAS 41, para. 3).

The relative importance of agriculture in the global economy has traditionally received little recognition, and therefore accounting in this field did not begin to attract attention from scholars in the accounting field until the issuance of IAS 41 by the IASB (Fischer and Marsh 2013; Herbohn and Herbohn 2006).

With the introduction of IAS 41 in 2001, accounting for agricultural activities became more flexible with the opportunity to switch from the cost to the fair value (FV) approach. In this context, agricultural enterprises have been given the choice of using the principle of acquisition cost minus accumulated depreciation and loss due to impairment (cost model), or the revaluation model (in which the fair value of can be reliably measured) to track variations in asset values. The resultant change in asset value influences the firm value (Ogilvy 2015). Since then, accounting scholars have discussed the pros and cons of FV accounting over historical costs (Bozzolan et al. 2016).

Alternatively, Filho et al. (2013) found that the adoption of fair value had a positive impact on the equity of the companies they analyzed, which in turn benefited all stakeholders. Thus, important perspectives can be noticed in terms of the decision maker's power in valuing biological assets, such as the application of the present value by measuring the discounted cash flow. The existence of specific parameters for determining discount rates, which are guided by recognized accounting standards may decrease the need for managerial judgment, and reduce the likelihood of errors and manipulation problems as well as enhancing comparability (Eckel et al. 2003).

This major switch from the traditional acquisition cost model to the FV model (Lefter and Roman 2007) caused an important debate on accounting within agriculture (Argilés et al. 2011). Subjectivity is a manifest challenge of FV due to the difficulty of computing FV, particularly if the market is not efficient and there is an absence or unavailability of Level 1 inputs for assets and liabilities (Pandya et al. 2021).

Within the content of IAS 41, accounting rules allow management discretion in selecting accounting practices, and this change has prompted research interest. For example, studies by Cormier et al. (2009) in France and Hellman (2011) in Sweden provided some signs regarding the implementation of discretionary power when applying the International Financial Reporting Standards (IFRS). Such discretion was evidenced in the use of the FV model in the absence of observable data, and results in managers being held responsible for using the present value method, which can make them more or less prudent in their forecasts (Silva et al. 2015).

Moreover, during the consultation on the IASB 2011 agenda, numerous respondents expressed concern about the cost complications and practical difficulty of measuring BA at FV, especially in the absence of an active market (Bozzolan et al. 2016).

Furthermore, confusion may occur due to the increased focus on the value relevance of accounting information at the expense of either cost reduction or discretionality, and the increased faithful representation of information secured by using the cost model. The best accounting practices might be governed by certain firm characteristics, whereas the suitable measurement and the pertinent disclosures of agricultural activities may vary across such characteristics (Gonçalves and Lopes 2015).

The relative importance of agriculture in the global economy has traditionally received little recognition, and therefore accounting in this field did not begin to attract attention from scholars in the accounting field until the issuance of IAS 41 by the International Accounting Standards board (IASB) (Fischer and Marsh 2013; Herbohn and Herbohn 2006).

The use of fair value to measure biological assets permits managerial judgment that may precipitate arbitrary decisions, particularly when using discounted cash flows and/or where an active market is absent, thereby affecting the amount of relevant information presented (Silva et al. 2015). However, Silva et al. (2015) were unable to find sufficient evidence of the level of discretionary differences; consequently, it is deemed valuable to inspect the accounting practices associated with measurement and disclosure within agricultural activities companies, and to do so using firm characteristics as variables. Additionally, as agricultural commodities represent a major portion of many developing countries' GDP including Jordan, there is merit in this study. Particularly, recent support and guidance are directed toward this sector in Jordan. Furthermore, the study findings provide key inputs to the agricultural sector in Jordan and other countries, enabling better management of resources at both the company and global levels. Jordan can be considered a good example to represent developing countries due to its early adoption of IAS.

Moreover, the measurement and disclosure practices pertaining to agricultural activities in Jordan need to be clarified and explored to facilitate the missions of responsible bodies in Jordan regarding the directions and financial or logistic supports. Exploring the accounting practices of agricultural firms based on general firm characteristics simplifies the direction of either guidance or financial and logistic supports according to these characteristics. These general characteristics are directly related to the companies, not to the board and management who ruled these companies. The general characteristics have a more constant nature compared to other characteristics associated with corporate board characteristics. A constant nature is more desirable for directing support and guidance.

To sum up, the current study does not aim to examine the pros and cons of FV versus historical cost, but rather focuses on exploring accounting practices related to measurement and disclosures in Jordanian companies that engage in agricultural activities based on their characteristics, and the extent to which companies adhere to the disclosure requirements mandated in IAS 41. Research in this area is worthwhile as it should be of value to standards setters in enhancing their understanding of measurement and disclosure practices with agricultural companies in a developing country.

## 2. Literature Review

### 2.1. The Conceptual Framework

IAS 41 regulates measurement, recognition, presentation and disclosure with respect to agricultural activities. This includes management of the biological change and the harvest produced by biological assets. The standard defines the treatment for those assets throughout their growth, degradation, production processes and reproduction, as well as for the initial recognition for either biological assets or of products at the time of harvest. It does not involve post-harvest transactions, as for instance, the processes of converting grapes into wine. IAS 41 includes the following (IFRS 2021b):

- *"bearer plants are accounted for using IAS 16;*
- *other biological assets are measured at fair value less costs to sell;*
- *agricultural produce at the point of harvest is also measured at fair value less costs*
- *changes in the fair value of biological assets are included in profit or loss; and*

- *biological assets attached to land (for example, trees in a plantation forest) are measured separately from the land."*

The fair value of a biological asset or agricultural produce is its market price less any costs to sell the product. Costs to sell include commissions, levies, and transfer taxes and duties (IFRS 2021b).

However, if the fair value cannot be measured reliably, companies can utilize cost measurement as a substitute. Determining fair value effectively is achieved by referring to the quoted price, providing that there is an active market. If the latter condition is not met, companies can refer to the most recent market transactions or sector benchmarks to determine the fair value or refer to the present value as a last choice (NZICA 2009).

IAS 41 differs from IAS.20 in terms of recognizing governmental grants. Grants that are connected to biological assets and are unconditional must be measured at fair value minus the cost of sale and be recognized in profit or loss when it is due. However, conditional grants are only recognized in the profit or loss account where the conditions attached to them are met (IFRS 2021b).

The International Accounting Standards Committee (IASC) issued IAS 41 in February 2001. In 2003, the IASB passed a revised IAS 41 as a part of its primary technical project agenda (IFRS 2021b).

Prior to 2005, BA were generally measured at acquisition cost. Harvested agricultural products were treated as inventories and "*measured at the lower of cost and net realisable value*" (Cairns et al. 2011; IFRS 2021c, IAS 2 para. 9).

Thereafter, the IASB launched its Agenda consultation in 2011, and several respondents noted that the usage of matured BA such as oil palms or rubber trees was similar to manufacturing, and hence, should allow for cost modelling, which meets the requirements of IAS.16. The cost complexity, and the difficulty of measuring the FV of bearer plants were also of concern to numerous respondents, especially in the absence of an active market for those assets (Bozzolan et al. 2016).

Subsequently, in 2013, and in response to several limitations raised in the consultation period, the IASB issued the Exposure Draft that was mainly associated with bearer plants. In this connection, in 2014, the IASB revised the scope of IAS.16 to embrace bearer plants associated with agricultural activities, that were formerly covered by IAS 41. Nevertheless, IAS 41 applies to products grown on these plants (IFRS 2021b).

Moreover, several Standards that consequently implied essential amendments to IAS 41, changed. These included IFRS 13 "Fair Value Measurement" issued in 2011, IFRS 16 "Leases" issued in 2016, modifications pertaining to the references to the "Conceptual Framework" in 2018, and improvements to the IFRS that were made annually up to 2020 (IFRS 2021b).

With respect to IAS/IFRS, IAS 41 was affected by the application of fair value. In this regard, Aryanto (2011) claimed that the effect of applying IAS 41 was not as positive as anticipated, an outcome which altered companies' financial comparability.

This radical shift from the historical cost model (Lefter and Roman 2007) has been the source of the debate over agricultural accounting (Argilés et al. 2011). "*IAS.41 has been criticised for being too academic and for introducing inappropriate measurement methods for biological assets*" (Herbohn and Herbohn 2006, p. 179). Indeed, the effective measurement of fair value is not easy to accomplish due to a variety of factors including lack of active market; difficulty in recognizing the characteristics of a bearer plant; the costs of adopting fair value exceeding the expected benefits; and revenue volatility (Aryanto 2011; Muhammad and Ghani 2014).

Moreover, Silva et al. (2012) found that some items such as management risks, are not disclosed. They also point to the difficulties inherent in the decision-making process in the context of Brazilian firms due to the use of fair value as a measurement for biological assets. Using the cost model was deemed by these researchers as more reliable, and unbiased.

Filho et al. (2013) observed that most Brazilian companies operating in the agriculture-food sector measure their biological assets using the fair value approach, but do not disclose

the method adopted when computing the fair value. As a result, it can be argued that the comparability is impaired, the relevance of information to users is reduced, and more space exists for undesirable earnings management. In this respect, Oliveira et al. (2015) suggested using the net present value to compute fair value, given the inconsistency of market values of dairy productions animals, which minimizes the comparability of financial information.

The costs of using fair value as an approach have also attracted attention, in which respect Elad and Herbohn (2011) conducted a study in three advanced countries namely: Australia, the United Kingdom, and France. This revealed the costs of adopting fair value to outweigh the expected benefits. Moreover, it was seen that disclosure practices lacked comparability, French companies tending not to disclose the required information about biological assets (Elad and Herbohn 2011). Furthermore, Zhang et al. (2019) asserted that firms implement the fair value model in order to cover their poor performance. Additionally, Oyewo et al. (2020, p. 51) found in their study that there was a "*tendency for managers to manipulate earnings owing to the inability of auditor to effectively test fair value estimates*". Hence, more effort is demanded in testing the estimates of fair value, and this ultimately implies additional fees (Alqatamin and Ezeani 2020).

In the context of New Zealand, the main problem associated with IAS 41 is the presentation of the unrealized gain and loss through the profit and loss account (NZICA 2009). However, other studies indicate the opposite viewpoint. Argilés et al. (2011), for example, found no significant difference between measuring biological assets at either cost or fair value when measuring future cash flows, although the results show greater predictability for future earnings based on the fair value paradigm. These scholars discovered several errors pertaining to the practices of the cost model implemented by Spanish farms. Furthermore, by interviewing Spanish stakeholders such as farmers and accountants, Argilés Bosch et al. (2012) found that respondents made major calculation errors and exhibited worse judgment when relying on the cost model rather than the fair value model. Hence, they concluded fair value to be more user-friendly in financial reporting and to bring improvements in decision-making, an outcome supported by Hadiyanto et al. (2018) on the grounds that firms adopting fair value present more reliable and relevant information than those using cost measurement (Hadiyanto et al. 2018).

Earnings information is particularly relevant for companies using the fair value model when measuring in-exchange biological assets since the approach might introduce less relevant information if companies measure their in-use biological assets by this means (Huffman 2018). In other words, fair value is more relevant for in-exchange assets than for in-use assets (Botosan and Huffman 2015; Marshall and Lennard 2016). Christensen and Nikolaev (2013) note that companies select the historical cost model for measuring either intangibles or property, plant and equipment (i.e., in-use assets).

Nonetheless, using fair value to measure a biological asset may allow for arbitrary discretion, particularly when using discounted cash flows in the absence of active markets, since this influences the value and quality of information (Silva et al. 2015). Furthermore, the cost of debt has been found to be greater for firms that use fair value to measure their biological assets in comparison with firms adopting the cost method (Daly and Skaife 2016). Islamic financial organizations might encounter burdens in presenting faithful data based on fair value (Marzuki et al. 2021). In this respect, Filho et al. (2013) have warned of the probability of subjectivity and undesirable earnings management practices as a result of using fair value. Likewise, subjectivity is seen as a main problem due to the difficulty of computing fair values, especially where the market is not efficient, and/or where Level 1 inputs for all assets and liabilities are unavailable (Pandya et al. 2021). These problems have required countries to establish institutional structures to simplify and support the valuation of assets and liabilities that are measured using the fair value approach (Oyewo 2020).

*2.2. Empirical Literature and Hypothesis Development*

Due to the limited literature concerning accounting practices for biological assets, the current study refers to work undertaken with other non-financial assets, such as investment

property (see Christensen and Nikolaev 2013; Quagli and Avallone 2010; Taplin et al. 2014; Daniel et al. 2010), and plant, property and equipment (Hlaing and Pourjalali 2012) as well as BA (Gonçalves and Lopes 2015; Scherch et al. 2013). It covers some factors related to firm characteristics, which are hypothesized to have an impact on BA measurement and level of disclosures, such as: intensity of BA, level of international activities, and audit for the Big Four.

**Intensity of BA:** Empirical evidence demonstrates that although many companies measure BA according to the FV principle, others reject the assumption of FV reliability and therefore, measure them based on cost. A recent study conducted by Rahman and Hossain (2020) has shown FV decisions to be influnced by the intensity of fixed assets alongside other factors. Moreover, Christensen and Nikolaev (2013), and Hlaing and Pourjalali (2012) pointed toward the influence of non-financial asset intensity on FV adoption as a measurement for property, plant and equipment. Similarly, the level of disclosure has been seen to increase with the upturn of the BA intensity (Scherch et al. 2013). Referring to stakeholder theory, Silva et al. (2012) advocated for a sufficient level of disclosure regulated by IAS 41, to ensure the provision of relevant information to stakeholders. This recommendation was particularly aimed at companies possessing a substantial amount of BA. Indeed, the level of disclosure based on IAS 41 has been found to be positively influenced by the intensity of BA (Gonçalves and Lopes 2015).

Given the literature reviewed, two hypotheses are developed regarding the intensity of biological assets as follows:

- The Jordanian companies engaged in agricultural activities with a higher intensity of BA are more likely to measure their BA based on FV rather than the cost model.
- The Jordanian companies engaged in agricultural activities with a higher intensity of BA are more likely to disclose more information based on IAS 41.

**Size:** Large firms report higher agency costs (Jensen and Meckling 1976), and *"have both the available resources and necessary incentives to comply with accounting standards"* (Cairns et al. 2011, p. 7). Daniel et al. (2010) offered two conflicting views on the influence of enterprise size on FV adoption. Although small businesses are expected to be more cautious about choosing FV due to the high implicit or potential costs, they might tend to use FV to reduce information asymmetry between investors and management. Quagli and Avallone (2010) also agreed that using size as an indicator of policy costs decreases the probability of implementing FV when revaluing investment property. Likewise, Rahman and Hossain (2020) have recently reported the FV revaluation decision to be influenced by firm size alongside other factors. Equally important is the observation by Gonçalves and Lopes (2015) that the size factor positively impacts upon the level of disclosure. Additionally, again, more recently, size as one of other structured-related variables has been reported to positively influence the level of disclosure, in comparison to performance variables that do not have the same impact on disclosure (Haddad et al. 2020). Glaum et al. (2013) referred to the greater volume of resources devoted to the accounting process that bigger firms possess, noting that this eventually leads to a higher reporting quality than that achievable by small firms, a conclusion supported by Depoers (2000), who noted that large firms must assure a sufficient level of disclosure for investment purposes. Hence, it is seen that the costs attributed to the high level of disclosure can be effectively justified by large companies. Whilst the literature does contain mixed empirical evidence for the role of size, in the current study, a positive sign is presumed for the relationship.

Therefore, two hypotheses are developed regarding firm size as follows:

- The Jordanian firms engaged in agricultural activities with a higher amount of total assets are more likely to measure their BA based on FV rather than the cost model.
- The Jordanian firms engaged in agricultural activities with a higher amount of total assets are more likely to disclose more information based on IAS 41.

**International activities:** in this respect, Daniel et al. (2010) found that companies with higher global operations are more likely to implement the FV approach, and Taplin et al.

(2014) established that Chinese companies listed on overseas stock exchanges are anticipated to adopt FV for non-financial assets. Moreover, the disclosure level of the firm is also positively correlated with its level of international activities (Daske et al. 2013). International trading activities inevitably entail the need for greater amounts of disclosed information, which in turn serve to assist firms in expressing their international position to all interested stakeholders (Oliveira et al. 2006).

Therefore, two hypotheses are developed regarding the level of international activities as follows:

- The Jordanian firms engaged in agricultural activities with a higher level of international activities are more likely to measure their BA based on FV rather than cost.
- The Jordanian firms engaged in agricultural activities with a higher level of international activities are more likely to disclose more information based on IAS 41.

**Audit for the Big Four:** Agency theory holds that agency costs are minimized when the financial report is audited by independent auditors (Jensen and Meckling 1976), since this reduces information asymmetry. That said, lower information asymmetry is also correlated with FV measurement (Fontes et al. 2018), but in spite of auditing efforts, there is a considerable risk of management bias in terms of reporting FV (Nordlund et al. 2022). This represents a predominant challenge to auditors who may be faced with difficulties in trying to efficiently investigate the estimates provided by managers who attempt to manipulate earnings (Oyewo et al. 2020). Such difficulties and the need to optimize the outcomes of auditors' efforts implies higher audit fees in testing FV estimates (Alqatamin and Ezeani 2020). With respect to the disclosure level, previous studies indicate a positive relationship between this and the financial report being audited by one of the Big Four (Glaum et al. 2013; Hodgdon et al. 2009). Additionally, the perceived quality of audit work and value relevance of accounting information are positively associated with the size of the audit firm (DeAngelo 1981; Abdollahi et al. 2020).

Therefore, two hypotheses are developed regarding the audit for the Big Four (whether the financial report is audited by one of the Big Four firms or not) as follows:

- The Jordanian companies engaged in agricultural activities that are clients with the Big Four firms are more likely to measure their BA based on FV rather than the cost model.
- The Jordanian companies engaged in agricultural activities that are clients with the Big Four firms are more likely to disclose more information based on IAS 41.

Studies conducted before the last amendment made on IAS 41 in 2014 mainly addressed measurement issues and their impact, and implied trade-offs between FV and cost. Furthermore, they were relatively limited in their attention to topics related to the comparison of the accounting practices pertaining to agriculture (including measurement, disclosure) based on firm characteristics. This is seen particularly after the last amendment made to IAS 41 and on other related and overlapping standards such as IFRS.13 and IFRS.16.

Consequently, this study focuses on rectifying that gap by describing the accounting practices of agricultural activities based on Jordanian firms' characteristics. Additionally, it covers all accounting practices within the content of IAS 41. Moreover, some items within the disclosure index have been developed to reflect the recent amendments made to IAS 41. In investigating the current practice, this study helps standards setters to understand the determinants of measurement and disclosure practices regarding agricultural activities within the context of Jordan.

Whilst the literature does contain mixed empirical evidence for the role of mentioned variables, in the current study, a positive sign is presumed for the relationships. Therefore, as mentioned above, eight hypotheses are developed regarding the influence of companies' characteristics on measurement practices and level of disclosures, which were as follows:

Hypotheses related to the impact of companies' characteristics on measurement practices.

**H1.** *The Jordanian companies engaged in agricultural activities with a higher intensity of BA are more likely to measure their BA based on FV rather than the cost model.*

**H2.** *The Jordanian firms engaged in agricultural activities with a higher amount of total assets are more likely to measure their BA based on FV rather than the cost model.*

**H3.** *The Jordanian firms engaged in agricultural activities with a higher level of international activities are more likely to measure their BA based on FV rather than the cost model.*

**H4.** *The Jordanian companies engaged in agricultural activities that are clients with the Big Four firms are more likely to measure their BA based on FV rather than the cost model.*

Hypotheses related to the impact of companies' characteristics on level of disclosures.

**H5.** *The Jordanian companies engaged in agricultural activities with a higher intensity of BA are more likely to disclose more information based on IAS 41.*

**H6.** *The Jordanian firms engaged in agricultural activities with a higher amount of total assets are more likely to disclose more information based on IAS 41.*

**H7.** *The Jordanian firms engaged in agricultural activities with a higher level of international activities are more likely to disclose more information based on IAS 41.*

**H8.** *The Jordanian companies engaged in agricultural activities that are clients with the Big Four firms are more likely to disclose more information based on IAS 41.*

Figure 1 shows the research model used to pursue the above hypothesized relationships between continuous variables.

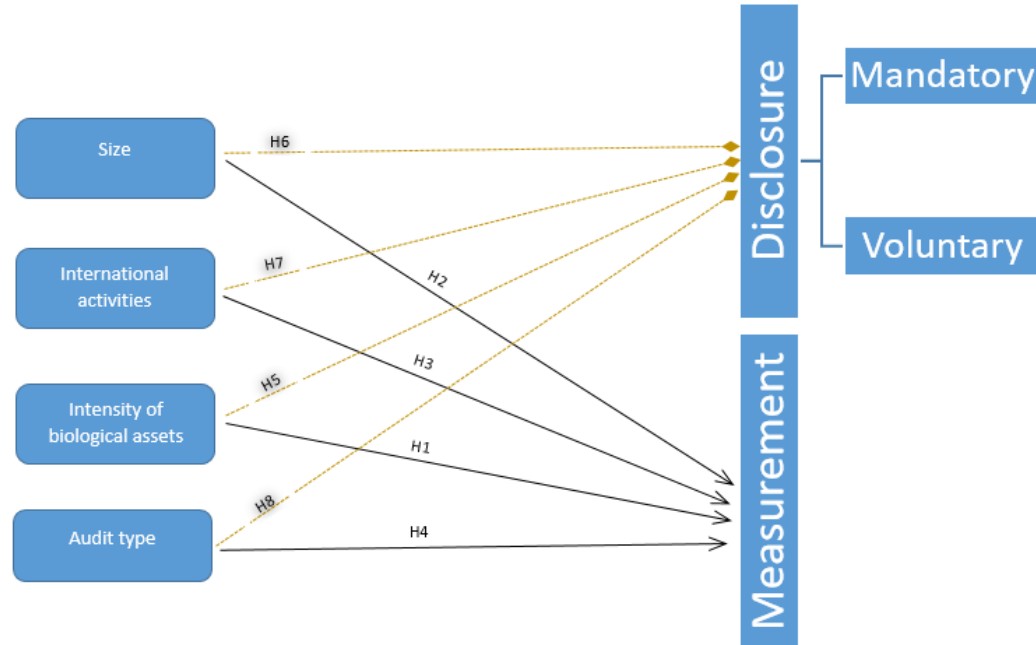

**Figure 1.** The Research Model.

### 3. Methodology

*3.1. Data and Sample*

As the Jordanian Companies Control Department allows companies to declare more than one purpose, the study population consisted of all reporting entities operating in Jordan that embraced one or more agricultural activity within its purposes. Reporting

requirements in Jordan mandate that all companies enjoying limited liabilities such as corporations and limited private companies prepare and publish their financial reports according to full IFRS and submit these reports to the Companies Control Department (SDC 2006). Companies that do not enjoy limited liabilities, such as general partnerships and limited partnerships, are only required by the Companies Law to prepare and publish their financial reports according to full IFRS and submit these reports to the Companies Control Department if their capital or turnover exceeds 100,000 JD (SDC 2006). Therefore, the population consisted of reporting entities with capital exceeding 100,000 JD and declaring agricultural activity within its purposes.

The Articles number based on the legal form that required this in the Jordanian companies' law are listed as the following:

- General partnership: Article 24 B.
- Limited partnership: Article 48.
- Limited liabilities: Article 62.
- Limited private company: Article 75 bis.
- Listed corporation: Articles 184–185 bis.

Although the regulator obliges all companies with limited liabilities to prepare and publish their financial reports according to full IFRS regardless of either capital or turnover amount, this study used the 100,000 JD as a minimum amount for the capital of companies included in its population. This was performed to eliminate micro or small entities since they usually encounter obstacles in adopting international standards when compared to medium and large entities. Additionally, their elimination was believed to enhance the consistency of sample, and to avoid size outliers. Equally important is the fact that the reporting entities followed the requirements of the Jordan Securities Commission regarding the disclosure instructions that were issued in 2004 and amended in 2019 (JSC 2019). Article 14 of disclosure instructions issued by the Jordan Securities Commission requires companies to prepare their financial information based on full IFRS in terms of presentation, measurement, and recognition (JSC 2019). The disclosures are directed by the Jordan Securities Commission.

After filtering the registered companies in the Companies Control Department based on the above criteria, the population of companies totaled 276 companies (that also replied to questionnaire), all of which were targeted for gathering secondary data via the disclosure index. However, 17 companies were excluded based on IAS 41 paragraphs 1, 5, 10, 12, and 30, that specified the companies that must be excluded from the adoption of IAS 41. Accordingly, the total number of population is 259 companies that were entirely targeted.

### 3.2. The Method

Given the dimensions of this study, a multimethod approach (see Tashakkori and Teddlie 2003) was used involving two quantitative instruments, these being a questionnaire survey and a disclosure index.

The structured questionnaire survey with financial managers was implemented to obtain information that could not be directly gathered from the financial reports of companies. In detail, the questionnaire survey (Appendix A) comprised fourteen questions about topics not featured in financial reports. These questions were divided into two main categories.

The first category contained five questions relating to inclusion and exclusion that were developed based on IAS 41 paragraphs 1, 5, 10, 12, and 30. If the answer to any of these questions was YES (except for Question number two where the answer should be NO), the company was excluded from the data set as this would mean that its agricultural transactions would be exempted from the requirements of IAS 41.

The second category contained nine questions associated with classification matters, which determined the number of items (in the disclosure index) that each company could disclose. For instance, Question 6 helped in classifying the company based on measurement, dividing companies into two categories, which eventually helped in specifying the number

of items within the index that each company could disclose according to the adopted measurement method. These questions were developed based on the paragraphs within IAS 41 and distributed to the financial managers of the targeted companies.

During the second half of 2022, the questionnaires were distributed and collected by the author in person or via email, or by filling the responses from financial managers by telephone. The contact information about companies were available at Jordanian companies control department as well as on the website of some companies. The questionnaires were distributed to 334 companies (all reporting firms engaged in agricultural activities), only 83% (276 questionnaires from 276 companies) were returned. However, 17 companies were excluded for the aforementioned reason. Thus, the final number of targeted companies was 259. The high response rate is due to that the questionnaire was designed to obtain factual responses related to adoption of IAS 41 rather than seeking opinions or perceptions. In addition, the number of question was not many (only 14 questions).

As shown in Appendix A, the index also includes the company number and was used to obtain data from financial reports. This comprised two main parts: the first pertaining to company characteristics, and the second associated with disclosure requirements (based on IAS 41 paragraphs). The amendments made to the indexes used in previous studies are shown in Appendix B together with the justifications for those changes. The number of paragraphs that were used in developing the disclosure index is provided for each items as illustrated in Appendices A and B.

In order to link the responses secured by these two techniques, the questionnaire for each company was cross-coded according to the published information before distribution, and the eventual questionnaire data were subsequently attached to the corresponding published data for each company. The respondents were informed of this process.

The questions regarding classification in the questionnaire were linked to specific items within the disclosure index (e.g., Question 9 about government grants linked with items 35–37 on the same topic) in order to specify the number of items that can be disclosed by the pertinent company, this number being used to compute the item index for each company. Both instruments were used to test the relationships between the variables accurately.

Table 1 presents the rationale for each question in the questionnaire (developed based on IAS 41 paragraphs).

The disclosure index measured the level of disclosure as the dependent variable, as has commonly been used in previous studies (Lopes and Rodrigues 2007; Oliveira et al. 2006; Akhtaruddin 2005; Owusu-Ansah 1998; Inchausti 1997). In terms of BA, Scherch et al. (2013) and Silva et al. (2012) used a disclosure index developed for the Brazilian context according to the disclosure requirements of IAS 41. That particular index is comprised of three main categories namely: mandatory disclosures; non-mandatory but recommended disclosures; and non-mandatory disclosures that are not recommended. The first and the second classifications cover all disclosure items required by the IAS 41, whereas the third category represents voluntary information which is only relevant to companies that measure their BA based on FV. However, Lopes and Rodrigues (2007) note that the disclosure index is unweighted and in their study, was adjusted for non-applicable items. Additionally, the use of an adjusted index regarding the disclosure level disregards the influence of the measurement method of BA on number of items in the index that each company can disclose.

Moreover, the index used in previous studies does not reflect the current update to IAS 41 made in 2014, nor other annual amendments.

To overcome this issue and the one pertaining to missing data, companies were classified according to their measurement methods as indicated in the questionnaire. As shown in Table 1, the questions regarding classification in the questionnaire are linked to specific items within the disclosure index (e.g., Question 9 about government grant in the questionnaire is linked with items 35–37 about government grant in the disclosure index) in order to specify the number of items that can be disclosed by the pertinent company as this number is used to compute each company's item index.

The disclosure index consisted of three questions and 37 items. The items concerning the disclosure practices were divided into two categories, namely mandatory and voluntary. Unlike in the previous disclosure index where voluntary items were only associated with the FV method, in this disclosure index some voluntary items were related to both the FV and cost methods.

Furthermore, the disclosure index in the current study contained amendments to the index used in previous research by for example, Scherch et al. (2013), and Silva et al. (2012), to reflect the current update made on IAS 41 and to suit the particularity of the context in terms of the regulation and nature of companies. The amendments made to the previous disclosure index are illustrated in Appendix B. Therefore, this study mainly referred to the IAS 41 to develop and update the disclosure indices used by previous studies.

**Table 1.** Questions in the First Research Instrument (Questionnaire).

| Question No | Rationale for Use |
|---|---|
| 1 | To exclude the company from the data set if the answer is yes. |
| 2 | To exclude the company from the data set if the answer is no. |
| 3 | To exclude the company from the data set if the answer is yes. |
| 4 | To exclude the company from the data set if the answer is yes. |
| 5 | To exclude the company from the data set if the answer is yes. |
| 6 | To classify companies based on measurement and specify the relevant companies for specific disclosure practice based on measurement method. This is considered to specify the number of items the company can disclose based on the measurement method. For instance, companies using the cost model cannot disclose some items included in the index such as items from 32 to 37. |
| 7 | To identify companies that previously measured BA based on cost. Only companies that answer yes are considered for item numbers 32–34 in the disclosure index (see Appendix A). |
| 8 | To classify companies based on FV measurement practices. Only companies measuring their BA based on FV can answer this question. |
| 9 | To classify companies according to whether they are entitled to a government grant. Only companies that have received a government grant are considered for item numbers 35–37 in the disclosure index (see Appendix A). |
| 10 | To identify whether the companies account its grant based on IAS.20 or IAS 41. in order to determine the practice for companies that are entitled for grant based on the answer of question 9. |
| 11 | To classify companies according to whether their financial statements are translated into foreign currency. Only companies that have performed this are considered for item number 17 in the disclosure index (see Appendix A). |
| 12 | To classify companies according to whether they are engaged in combination. Only companies that are engaged in combination are considered for item number 16 in the disclosure index (see Appendix A). |
| 13 | To classify companies according to whether they pledge BA as security for liabilities. Only companies that do this are considered for item number 9 in the disclosure index (see Appendix A). |
| 14 | To classify companies according to whether they are involved in international activities and to determine the effect of those activities on measurement practices and the level of disclosure. |

The details of the three questions related to company characteristics as well as the 37 disclosure items are shown in Table 2.

The entire disclosure level was computed by aggregating the score of all disclosed items, giving a score of 1 to an item if it was disclosed, and a score of 0 if it was not disclosed. Accordingly, the total score of the disclosure index is:

$$item\ index_i = \frac{\sum_{i=1}^{n} d_i}{n}$$

where $d_i$ equals 1 if the item is disclosed or 0 if the items is not disclosed; $n$ is the number of items that a firm can disclose. The questions in the questionnaire pertaining to classifications are used to determine the ($n$) for each company, thereby enhancing the adequacy of results.

Measurement practices is measured according to measurement method, where it equals 1 if the company measure its BA based on FV and 0 if it is based on the cost method.

**Table 2.** Questions and Items Comprising the Second Research Instrument **(the Disclosure Index)**.

| Question or Item Number | Reference | Measurement | Rationale for Use |
|---|---|---|---|
| Q1 | (Gonçalves and Lopes 2015; Scherch et al. 2013; Silva et al. 2012) | Amount of BA divided by amount of total assets | Used to find the impact of BA intensity on company practices. |
| Q2 | (Gonçalves and Lopes 2015; Glaum et al. 2013; Daniel et al. 2010; Quagli and Avallone 2010) | Natural logarithm of amount of total assets | Used to find the impact of size on company practices. |
| Q3 | (Gonçalves and Lopes 2015; Glaum et al. 2013; Hodgdon et al. 2009) | Dummy variable according to whether the company's financial report is audited by one of the Big Four auditing firms | Used to find the impact of audit for the Big Four on company practices. |
| Items 1–17 | IAS 41. para 40, 42, 46, 49, and 50. | Dummy variable | Represents the mandatory disclosure applicable to companies measuring their BA based on FV. Applicable to 104 companies. |
| Items 1–2, 4–11, and 13–27 | IAS 41. para 40, 42, 46, 49, and 54. | Dummy variable | Represents also the mandatory disclosure applicable to companies measuring their BA based on cost. Applicable to 155 companies. Except for:<br>• Item 9 applicable only to 26 companies (that pledge BA as security for liabilities).<br>• Item 16 applicable only to 9 companies (that engage in combination).<br>• Item 17 applicable only to 36 companies (that translate financial statements into foreign currency). |
| Items. 28–31 | IAS 41 Para 43 and 51 | Dummy variable | Represents the voluntary disclosure applicable to companies measuring their BA based on FV. Applicable to 104 companies. |
| Items 28–29 | IAS 41 Para 43 | Dummy variable | Represents also the voluntary disclosure applicable to companies measuring their BA based on cost. Applicable to 155 companies. |
| Items 32–34 | IAS 41 Para 56 | Dummy variable | Represents the mandatory disclosure applicable to companies measuring their BA based on FV but that previously measured based on cost. Applicable to 41 companies. |
| Items 35–37 | IAS 41 Para 57 | Dummy variable | Represents the mandatory disclosure applicable to companies measuring their BA based on FV and having a government grant. Applicable to 28 companies. There were another 24 companies entitled to a government grant but that measured their BA based on cost and then used IAS.20 to account for that grant. The latter group are not targeted for this set of items (34–37). |

## 4. Empirical Results

After applying the eligibility criteria related to IAS 41 paragraphs 1, 5, 10, 12, and 30, that specified the companies that must be excluded from the adoption of IAS 41, a total of 17 companies were excluded, as the BA of 15 companies were bearer plants. These bearer

plants are used in the production or supply of agricultural produce, produced for more than one period, or likely to be sold as agricultural produce. Five companies out of the 17 did not control their BA.

Given that some companies reported more than discarding issue related to questions from 1 to 5. The eventual population was 259 companies. That was entirely targeted in this study for collecting data via index.

Figure 2 shows the number and percentage of companies according to their measurement method. Approximately 60% (155 companies) used the cost method to account their BA, while the rest (104 companies) relied upon the FV method.

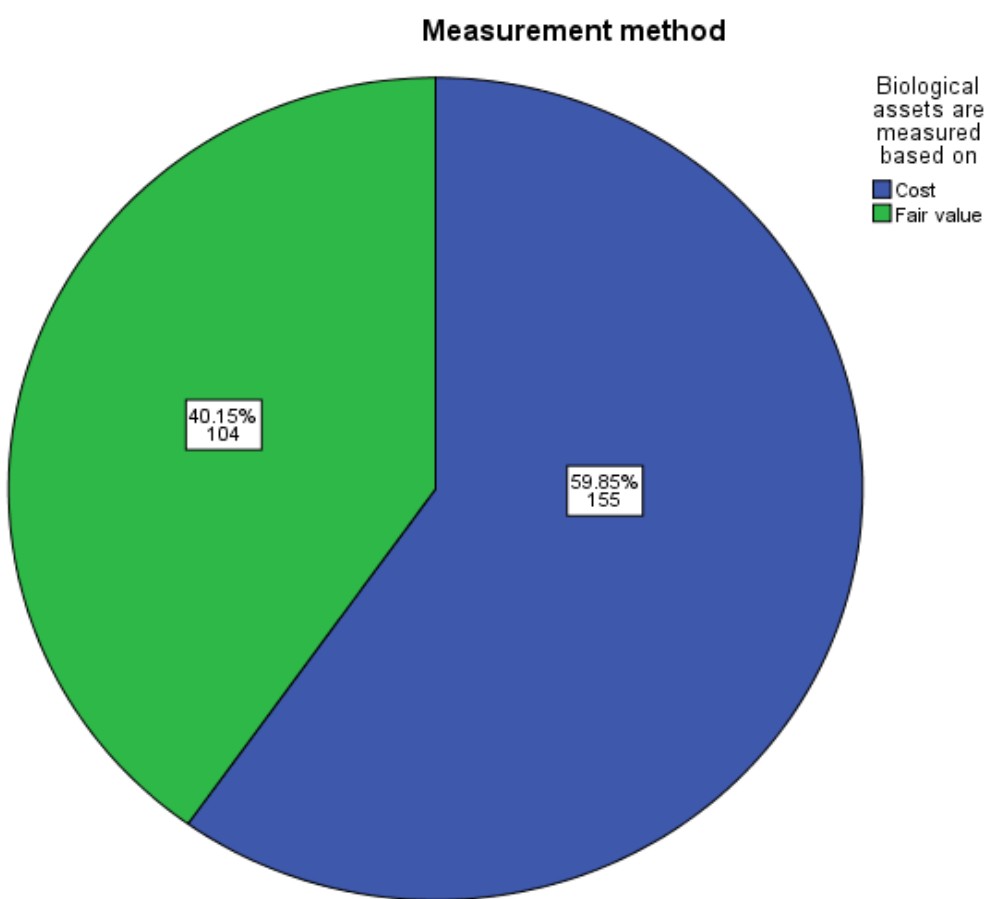

**Figure 2.** Number and Percentage of Companies According to Their Measurement Methods.

Only 39.4% (41) of companies that used FV (104 companies) for measuring their BA had previously used the cost method. The remaining 63 companies had adopted the FV approach since their commencement date of operations as illustrated in Figure 3.

When companies were classified as using FV as their method for measuring their BA, it was seen (Figure 4) that the present value and replacement cost methods were not used in computing the FV. While almost 35% (36 companies) used quoted prices for their BA in the active market, the remainder used the quoted prices for similar assets in either an active or non-active market because of the diversity of BA possessed by them. Quoted prices for similar assets in an active market was the most used method whilst quoted prices for similar assets in a non-active market was the least used method.

Table 3 present the cross-tabulated frequencies between the level of international activities and audit for the Big Four. The vast majority of the sampled companies (77.2%) did not audit their financial reports using one of the Big Four accounting firms. Approximately 31% of companies have a good or high level of international activities, while the rest have either a poor level or no international activities at all (30.1% and 38.6%, respectively).

Importantly, it is obvious that the number of companies whose financial reports were audited by one of the Big Four increases with the level of international activities. As demonstrated, only three companies of the 100 that had no international activities used a Big Four auditor, whereas the pattern reverses in terms of companies that report a high level of international activities, with 20 out of 31 companies reporting using a Big Four firm.

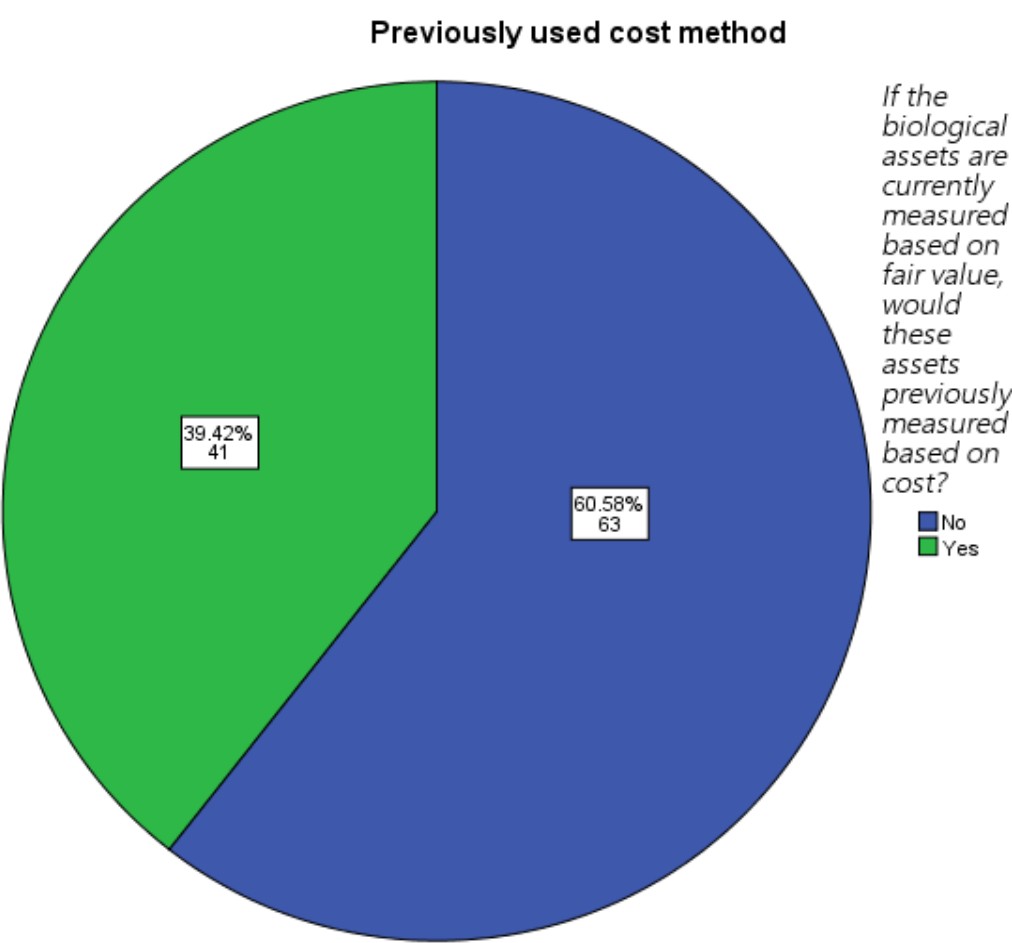

**Figure 3.** Number of Companies Previously Using/Not Using the Cost Method.

**Table 3.** Levels of International Activities and Audit for the Big Four.

| International Activities | | How Do You Rate the Level of International Activities of Your Company? | | | | Total | Percent |
|---|---|---|---|---|---|---|---|
| Audit for the Big Four | | There Are No International Activities | Poor | Good | High | | |
| Is the company financial report is audited by one of the Big Four audit firm? | No | 97 | 65 | 27 | 11 | 200 | 77.2% |
| | Yes | 3 | 13 | 23 | 20 | 59 | 23.8% |
| Total | | 100 | 78 | 50 | 31 | 259 | 100% |
| Percent | | 38.6% | 30.1% | 19.3% | 12% | 100% | |

**Figure 4.** Number and Percentage of Companies According to Methods Used for Measuring BA Using FV.

Table 4 shows some descriptive statistics regarding the intensity of BA and total assets. The mean scores indicate that the percent of BA to total assets is approximately 33% (993238.88/2974073.33). However, by reviewing the data set, the percentages of BA to total assets are seen to vary across size clusters, legal form and sector type.

**Table 4.** Descriptive Statistics—Intensity of BA and Size.

| Variables | Range | Minimum | Maximum | Mean |
|---|---|---|---|---|
| The Intensity of biological assets | 18,508,455 | 13,125 | 18,521,580 | 993,238.88 |
| Size—total assets | 49,835,700 | 210,300 | 50,046,000 | 2,974,073.33 |

The descriptive analysis presented in Table 5 reveals the overall disclosure level and mandatory disclosure level to be relatively low within Jordanian companies, these being 34.17% and 39.47%, respectively. One reason might be that some companies do not enjoy limited liabilities (limited partnership and partnership) and are therefore not mandated to provide that information, whereas those with limited liabilities (limited private company, corporation, and limited liabilities companies) are obligated to do so by the Jordanian Securities Commission rules. Overall, the voluntary disclosure level was very low because

Jordan does not have any institutional body requiring such disclosure. As aforementioned, 40.15% of companies use the FV measure.

**Table 5.** Descriptive Statistics of Study Variables.

| Variables | Mean | Std. Deviation |
|---|---|---|
| Overall disclosure level | 0.3417 | 0.1129 |
| Mandatory disclosure level | 0.3947 | 0.4231 |
| Voluntary disclosure level | 0.1372 | 0.1118 |
| Measurement practices | 0.4015 | 0.4911 |

Multiple regression is not appropriate when the outcome variable is categorical. Therefore, logistic regression is used to test the impact of firms' characteristics on measurement practices (H1 to H4) as the dependent variable is categorical, where the predictor can be categorical or continuous.

Testing the impact of firms' characteristics on disclosure level (H5 to H8) is performed by multiple regression as the independent variable is ratio. Given that two separate models were estimated for testing this impact, one for the companies that use FV to measure their BA and another for the companies that use the cost method.

With reference to Tabachnick and Fidell (1996), the problem of multicollinearity between the variables arises for correlation coefficients more than 90%. For other researchers such as Gujarati (1995), the correlations must not exceed 80% in order to avoid the multicollinearity problem. As illustrated in Table 6, the maximum correlation coefficient was 0.570, between audit for the Big Four and firm size, which indicates that the multicollinearity issue is not presented in the current study. The VIF and the tolerance are also tested and explained throughout regression analysis to ensure the absence of the multicollinearity problem.

**Table 6.** Correlation of Independent Variables.

| | International Activities | Intensity of BA | Audit | Size |
|---|---|---|---|---|
| International activities | 1.000 | | | |
| Intensity of BA | 0.046 | 1.000 | | |
| Audit for the Big Four | 0.077 | 0.122 | 1.000 | |
| Size | 0.080 | 0.253 | 0.570 | 1.000 |

**Model (1): Impact of firms' characteristics on measurement practices.**

- To estimate the effect of company characteristics on measurement practices (H1 to H4), the following equation for logistic regression of model (1) was used:

$$P(Measurment\ practice) = \frac{1}{1 + e^{-(\beta_0 + \beta_1 Intensity\ of\ BA + \beta_2 Size + \beta_3 International\ activities + \beta_4 Audit\ for\ BF)}} \tag{1}$$

The regression results of model (1) are shown in Tables 7 and 8, and explained below.

**Table 7.** Model Summary and the Hosmer and Lemeshow Test.

| | Model Summary | | | Hosmer and Lemeshow Test | | Omnibus Test | |
|---|---|---|---|---|---|---|---|
| Step | −2 Log Likelihood | Cox and Snell R Square | Nagelkerke R Square | Chi-Square | Sig. | Chi-Square | Sig. |
| 1 | 188.961 [a] | 0.461 | 0.623 | 11.354 | 0.182 | 133.188 | 0.000 |

[a] Estimation terminated at iteration number 7 because parameter estimates changed by less than 0.001.

**Table 8.** Coefficients for Company Characteristics on the Probability of Measuring BA by FV.

|  |  | **B** | **S.E.** | **Sig.** | **Odds Ratio** |
|---|---|---|---|---|---|
| Step 1 [a] | International activities | 1.659 | 0.247 | 0.000 | 5.256 |
|  | Intensity of BA | 0.000 | 0.000 | 0.003 | 1.000 |
|  | Audit for the Big Four | 0.199 | 0.650 | 0.760 | 1.220 |
|  | Size | 0.000 | 0.000 | 0.784 | 1.000 |
|  | Constant | −3.049 | 0.363 | 0.000 | 0.047 |

[a] Variable(s) entered on step 1: International, BA, audit, and size. Dependent variable: measurement practice is given a value of 1 if BA was measured by FV, and 0 if BA was measured by the cost method.

Logistic regression was executed to assess the impact of firms' characteristics on measurement practice. The model consisted of four independent variables (level of international activities, intensity of Bay, firm size, and audit for the Big Four). As shown in Table 7, this model was statistically significant (Omnibus test) $\chi^2$ (4, N = 259) = 133.188, $p < 0.001$, demonstrating that the model is able to distinct firms that measured their BA based on FV from those measured based on the cost method. The model explained between 47.1% (Cox and Snell R square) and 62.3% (Nagelkerke R squared) of the variance in measurement methods.

Regarding the Hosmer–Lemeshow Goodness of Fit, Table 7 showed that the significance value was more than 0.05, which supports this model, where the chi-square value was 11.354, with a significance level of 0.182.

As exhibited in Table 8, only two of the independent variables made a unique statistically significant contribution to the model (level of international activities, and intensity of BA). The strongest predictor of reporting a FV was the level of international activities with an odds ratio of 5.256. This indicated that firms with a higher level of international activities were 5.256-fold more likely to measure their BA based on FV.

**Model (2): Impact of firms' characteristics on disclosure level for firms measured their BA based on the FV and cost methods**

- To estimate the effect of company characteristics on disclosure level (H5 to H8), the following equation was used:

$$Disclosure\ level = \beta_0 + \beta_1 Intensity\ of\ BA + \beta_2 Size + \beta_3 International\ activities + \beta_4 Audit\ for\ BF + \varepsilon \quad (2)$$

The regression results of model (2) are shown in Tables 9 and 10 and explained below.

**Table 9.** Model Summary and ANOVA.

| **Model** | **Df** | **R** | **R Square** | **Adjusted R Square** | **F** | **Sig.** |
|---|---|---|---|---|---|---|
| Regression | 4 |  |  |  |  |  |
| Residual | 254 | 0.580 | 0.337 | 0.326 | 32.257 | 0.000 |
| Total | 258 |  |  |  |  |  |

Dependent variable: disclosure level.

Based on the R square score presented in Table 9, the model (which includes intensity of BA, size, international activities, and audit for the Big Four) explains 33.7% of the variance in overall disclosure level. ANOVA findings indicate that the model was statistically significant: F (4, 254) = 32.257, $p < 0.01$.

**Table 10.** Coefficients for Company Characteristics on Overall Disclosure (all firms).

| Model | Unstandardised Coefficients | | Standardised Coefficients | T | Sig. | Collinearity Statistics | |
|---|---|---|---|---|---|---|---|
| | B | Std. Error | Beta | | | Tolerance | VIF |
| (Constant) | 0.143 | 0.008 | | 17.354 | 0.000 | | |
| International activities | 0.014 | 0.007 | 0.131 | 2.117 | 0.035 | 0.682 | 1.465 |
| The intensity of BA | $-3.602 \times 10^{-9}$ | 0.000 | −0.079- | −0.605 | 0.546 | 0.526 | 1.902 |
| Size—total assets | $2.091 \times 10^{-9}$ | 0.000 | 0.133 | 0.972 | 0.332 | 0.578 | 1.729 |
| Audit for the Big Four | 0.125 | 0.017 | 0.463 | 7.139 | 0.000 | 0.620 | 1.614 |

Table 10 demonstrates that the Sig. values were less than 0.05 for both the international activities and audit for the Big Four variables, thereby indicating that these two variables make a significant contribution to the prediction of overall disclosure level, as they positively influence (due to beta signs) overall disclosure. On the contrary, the intensity of BA and size variables have Sig. values exceeding 0.05, demonstrating no significant impact on overall disclosure level made by either the intensity of BA or size variables.

The highest standardized coefficient of beta score was 0.463, which was for the audit for the Big Four variable. Therefore, audit for the Big Four makes the strongest contribution to explaining the overall disclosure level, thus implying that maximizing the scores of the audit for the Big Four variable by one standard deviation will result in increasing the overall disclosure level by 0.463 standard deviation units. The international activities variable also makes a statistically significant contribution (beta = 0.131).

The VIF values for all variables are less than 10, and the tolerance values exceed 0.10, indicating an absence of multicollinearity.

As illustrated in Table 11, the null hypothesis of homoscedasticity or constant variance is accepted because the *p*-value was more than 0.05 for the Breusch–Pagan/Cook–Weisberg test. This implied that there is no evidence of heteroscedasticity problem.

**Table 11.** Breusch–Pagan/Cook–Weisberg Test for Heteroskedasticity.

| Ho: Constant Variance | |
|---|---|
| chi2(1) | 0.07 |
| Prob > chi2 | 0.7981 |

As mentioned above, two separate models were estimated to test the influence of firms' characteristics on disclosure level, one for the companies that use FV and another for the companies that use cost.

The main reason for estimating these two separate models is that the results of the Chow test, F (5, 249) = 15.32, *p* < 0.01, as illustrated in Table 12, indicated that the behavior in terms of the impact on disclosure level, was differed between companies that measured its BA based on FV and those measured BA based on the cost method. Therefore, the conclusion regarding the effect of companies' characteristics on disclosure level based on IAS 41 can be drawn from the results of model (2a) (for companies' measured BA based on FV) and model (2b) (for companies' measured' BA based on the cost method) instead of relying upon the results of model (2) (for companies' measured BA by any measurement method).

**Table 12.** The Results of the Chow Test.

| Source | Sum of Squares | df | Mean Square | F | Sig. |
|---|---|---|---|---|---|
| Contrast | 0.527 | 5 | 0.105 | 15.832 | 0.000 |
| Error | 1.657 | 249 | 0.007 | | |

- To estimate the effect of company characteristics that measured BA based on FV on disclosure level, the following equation was used:

$$Disclosure\ level\ for\ firms\ use\ FV\ to\ measure\ BA = \beta_0 + \beta_1 Intensity\ of\ BA + \beta_2 Size + \beta_3 International\ activities + \beta_4 Audit\ for\ BF + \varepsilon \quad \text{(2a)}$$

- To estimate the effect of company characteristics that measured BA based on the cost method on disclosure level, the following equation was used:

$$Disclosure\ level\ for\ firms\ use\ cost\ to\ measure\ BA = \beta_0 + \beta_1 Intensity\ of\ BA + \beta_2 Size + \beta_3 International\ activities + \beta_4 Audit\ for\ BF + \varepsilon \quad \text{(2b)}$$

The regression results of models (2a) and (2b) are shown in Tables 13 and 14 and explained below.

**Table 13.** Models Summary and ANOVA.

| | Df | R | R Square | Adjusted R Square | F | Sig. |
|---|---|---|---|---|---|---|
| **Model (2a) (Firms Measured BA Based on FV)** | | | | | | |
| Regression | 4 | | | | | |
| Residual | 99 | 0.651 | 0.424 | 0.401 | 18.211 | 0.000 |
| Total | 103 | | | | | |
| **Model (2b) (firms measured BA based on cost)** | | | | | | |
| Regression | 4 | | | | | |
| Residual | 150 | 0.726 | 0.527 | 0.515 | 41.860 | 0.000 |
| Total | 154 | | | | | |

Dependent variable: disclosure level.

Based on the R square score presented in Table 13 for model (2a) (for firms measured BA based on FV), this explains 42.4% of the variance in overall disclosure level. ANOVA findings indicate that the model (2a) was statistically significant: F (4, 99) = 18.211, $p < 0.01$.

The R square score of model (2b) (for firms measured BA based on the cost method), which also includes intensity of BA, size, international activities, and audit for the Big Four, explains 52.7% of the variance in overall disclosure level. ANOVA findings indicate that model (2b) was statistically significant: F (4, 150) = 41.860, $p < 0.01$.

**Table 14.** Coefficients for Company Characteristics on Overall Disclosure.

| | **Model (2a)** | | | | **Model (2b)** | | | |
|---|---|---|---|---|---|---|---|---|
| | Std. Error | Standardised Coefficients Beta | T | Sig | Std. Error | Standardised Coefficients Beta | t | Sig |
| (Constant) | 0.021 | | 6.897 | 0.000 | 0.010 | | 10.444 | 0.000 |
| International activities | 0.010 | 0.418 | 5.697 | 0.000 | 0.030 | 0.380 | 5.413 | 0.000 |
| Intensity of BA | 0.000 | 0.068 | 0.335 | 0.738 | 0.000 | 0.273 | 4.396 | 0.000 |
| Size (total assets) | 0.000 | 0.108 | 0.524 | 0.602 | 0.000 | 0.027 | 0.360 | 0.719 |
| Audit for the Big Four | 0.020 | 0.563 | 6.341 | 0.000 | 0.009 | 0.410 | 7.086 | 0.000 |

Model (2a) in Table 14 reveals that the Sig. values were less than 0.05 for audit for the Big Four and the level of international activities variables, thereby showing that these variables make a significant contribution to the prediction of overall disclosure level for companies measuring their BA based on FV, as they positively influence (due to beta signs) overall disclosure. The highest standardized coefficient of beta score in model (2a) was 0.563, which was for audit for the Big Four. This implies that maximizing the scores of the audit for the Big Four variable by one standard deviation will result in increasing the overall disclosure level by 0.563 standard deviation units. On the contrary, the intensity of BA and size variables have Sig. values exceeding 0.05, demonstrating no significant impact of these variables on overall disclosure level.

Model (2b) in Table 14 demonstrates that the Sig. values were less than 0.05 for all variables except firm size, thereby indicating that these variables make a significant contribution to the prediction of overall disclosure level for the companies that measured their BA based on the cost method, as they positively influence (due to beta signs) overall disclosure.

The highest standardized coefficient of beta score in model (2b) was 0.410, which was for audit for the Big Four. Therefore, the latter variable makes the strongest contribution to explaining the overall disclosure level for the companies that measured their BA based on the cost method, thus implying that maximizing the scores of the audit for the Big Four variable by one standard deviation will result in increasing the overall disclosure level by 0.410 standard deviation units. This is followed by the level of international activities and then the intensity of BA, which also make a statistically significant contribution to predict the overall disclosure level.

Although the results regarding the variables affecting overall disclosure level for the companies that measured their BA by any measurement method model (2) were similar to those that measured their BA based on the FV model (2a), the results differed from those that measured their BA based on the cost method model (2b). Regarding the rank of variables that contribute in explaining the overall disclosure level, the audit for the Big Four variable has the strongest contribution to explaining the overall disclosure level for the two models (2a) and (2b), followed by the level of international activities variable. The intensity of the BA variable contributes significantly only in predicting the overall disclosure for companies that measure their BA based on the cost method model (2b), given that its influence was less than both the audit for the Big Four and level of international activities variables as demonstrated by the standardized coefficient of beta score.

## 5. Discussion

This study aimed to examine the impact of firms' characteristics on measurement and disclosure practices pertaining to biological assets based on the requirement prescribed by IAS 41 in the context of Jordanian companies that have agricultural activities. This study is mainly distinct from previous studies as it used the developed disclosure index to reflect the most updated amendments made in IAS 41 by the IASB, especially those conducted after 2014 as aforementioned. Further, this study investigated the measurement and disclosure practices related to agricultural activities and primarily to BA rather than other non-financial assets as most studies did. Furthermore, the adjusted index in the current study overcame the issue regarding the influence of the measurement method of BA on the number of items that the company can disclosed as already explained.

The findings of this study reveal that the intensity of BA has positive impact on measurement practices as indicated by the results of H1 that was accepted. The findings were alligned with those of other studies (e.g., Rahman and Hossain 2020; Christensen and Nikolaev 2013; Hlaing and Pourjalali 2012) that found an influence coming from non-financial asset intensity on the adoption of FV. This can be traced back to the fact that the companies would be encouraged to use the FV to measure their BA that represent a significant amount of total assets, as FV provides all stakeholders with more relevant

information about the market value of BA compared to the cost method. Particularly, BA is subject to change in value at a rate higher than other assets due to their nature.

However, the intensity of BA has no impact on the level of disclosure for companies that measure its BA based on the FV model (2a) that resulted in rejecting H5 based on model (2a). This finding contradicts those of other researchers (e.g., Scherch et al. 2013; Gonçalves and Lopes 2015). This can be justified based on the fact that the adoption of FV may impose further cost. Or reduce the relevance, as Filho et al. (2013) observed that most Brazilian companies operating in the agriculture-food sector that measure their biological assets using the fair value approach, do not disclose the method adopted when computing the fair value. This also justified by the results of model (2b), that indicated a positive impact on the level of disclosure for firms measuring their BA based on the cost method, which implied accepting the H5 based on the results of model (2b). To sum up the disclosure level was influenced by firms' characteristics that used the cost method for measuring BA due to the simple calculation of BA cost in comparison to the complexity related to measuring BA based on FV that may imply further cost pertaining to measure FV reliably and reduce the relevance (Filho et al. 2013).

Moreover, to the best of the researcher's knowledge, there has been limited attention toward the influence of intensity of BA on FV adoption since the amendment made to IAS 41 in 2014 regarding measurement requirements. Previous studies (e.g., Rahman and Hossain 2020; Christensen and Nikolaev 2013; Hlaing and Pourjalali 2012) tested the impact of non-financial asset intensity, whereas the current study has examined the impact of BA intensity. Therefore, various findings could be raised.

H2 and H6 relating to the positive impact of firm size on measurement practice and disclosure level were rejected. The results with respect to the influence on disclosure level were not in agreement with those achieved by other scholars such as Haddad et al. (2020) and Gonçalves and Lopes (2015). This can be traced back to the fact that the target group includes listed and non-listed entities whose disclosure practices vary according to company law and Jordanian Securities Commission requirements. This may also be the reason for the difference in findings between the current and previous studies in terms of the influence on measurement practices. The trend of small businesses in terms of their use of FV is not constant because they are cautious about the high costs associated with the approach, but they do appreciate that its use may help to decrease information asymmetry (Daniel et al. 2010). However, large firms report higher agency costs (Jensen and Meckling 1976), and *"have both the available resources and necessary incentives to comply with accounting standards"* (Cairns et al. 2011, p. 7). The findings were not consistent with those of Quagli and Avallone (2010) and Rahman and Hossain (2020), who tested the influnce of firm size on the revaluation decision. Moreover, most earlier studies tested the impact of firm size on the revaluation decision or using FV for non-finnancial assets rather than to measure BA as has been performed in the current study.

With respect to the level of international activities, H3 and H7 were accepted, confirming the positive impact of the level of international activities on measurement practices model (1) as well as on the level of disclosure models (2a) and (2b). Additionally, here, the results concurred with those of Daniel et al. (2010) and Taplin et al. (2014) concerning measurment practices. Likewise, the results were aligned with those of Daske et al. (2013). The reason for these outcomes is their association with the idea that companies are assisted in their efforts to convey their international position to stakeholders (Oliveira et al. 2006).

With regard to audit for the Big Four, H4 was rejected, thereby confirming the absence of impact of audit for the Big Four on measurement practices. This can be justified based on the existence of several challenges as highlighted by Nordlund et al. (2022), Alqatamin and Ezeani (2020), and Oyewo et al. (2020) relating to the risk of management bias, the difficulties faced by auditors in investigating the estimates of measured fair value, and increasing audit fees.

On the other hand, the findings of the current study show that audit for the Big Four positively impacts the overall disclosure level and makes the strongest contribution to

explaining the overall disclosure level for all models (2a) and (2b), agreeing with previous researchers (e.g., Glaum et al. 2013; Hodgdon et al. 2009). The quality of the audit work and the value relevance of accounting information are positively related to the size of the audit firm (DeAngelo 1981; Abdollahi et al. 2020). This is in addition to minimizing the agency cost by having the financial report audited by independent auditors (Jensen and Meckling 1976), leading to reduced information asymmetry.

## 6. Conclusions

The aim of this study is to describe the accounting practices of the agricultural activities of Jordanian firms based on those firms' characteristics, and was conducted in light of IAS 41 in terms of financial disclosure and measurement.

The findings revealed that both intensity of BA and level of international activities have a positive impact on measurement practices, where the contribution of the level of international activities variable better explained measurement practices compared to the intensity of BA variable. Audit for the Big Four was the strongest variable that positively influenced the overall disclosure level prescribed by IAS 41, followed by the level of international activities. The intensity of the BA variable affects only the overall disclosure level for companies that measure their BA based on the cost method model (2b). Firm size was found to have no influence on either measurement practices or disclosure level.

The results of this article may have theoretical and practical implications. This study provided new insights into the influence of firms' characteristics on measurement and disclosure practices for companies that include agricultural activities—particularly in the context of developing countries, such as Jordan. This contributed to the body of literature concerning either accounting standards related to agricultural activities in general or biological assets in particular.

The current study contributed to the literature of financial reporting by examining the influence of firms' characteristics on measurement and disclosure practices, where the previous studies essentially tested the impact of agricultural firms on disclosure level with evident absence of measurement practice. However, other previous studies investigated the impact on either measurement or disclosure that was related to non-financial assets rather than BA.

In terms of methodological contribution, the problems pertaining to the disclosure indices used in previous studies regarding disregarding the influence of the measurement method of BA on the number of items in the index that each company can disclose was overcome. This was overcome by using the question in the questionnaire pertaining to measurement of BA based on FV or cost in order to determine the number of items that each company can disclose, thereby enhancing the adequacy of results, given that two separate models were estimated to test the influence of firms' characteristics on disclosure level, one for the companies that use FV and another for the companies that use cost, which also enriches the findings and enhances accuracy.

Moreover, the index used in previous studies does not reflect the current update to IAS 41 made in 2014, nor other annual amendments. The index of current studies relied mainly upon IAS 41 and reflected the most updated items.

The items concerning the disclosure practices were divided into two categories, namely mandatory and voluntary. Unlike in the previous disclosure index, where voluntary items were only associated with the FV method, some voluntary items were related to both the FV and cost methods in this disclosure index.

Furthermore, the disclosure index in the current study contained amendments to the index used in previous research by, for example, Scherch et al. (2013) and Silva et al. (2012), to reflect the current update made on IAS 41 and to suit the particularity of the context in terms of the regulation and nature of companies. The amendments made to the previous disclosure index are illustrated in Appendix B. Therefore, this study mainly referred to IAS 41 to develop and update the disclosure indices used by previous studies.

Equally important, numerous practical implications can be directed toward management, standards setters and other stakeholders especially in developing countries. This study helps standards setters and regulators to understand the determinants of measurement and disclosure practices based on companies' characteristics within the context of Jordan, which in turn facilitates amending and directing financial regulation according to these determinants. This eventually promotes the accounting practices related to agricultural activities. To enhance this, establishment of an institutional structure is necessary to simplify and support the valuation of BA measured based on fair value. This would enhance comparability, reduce implicit costs as well as avoid the additional audit fees, eliminate or at least mitigate arbitrary discretionality and the difficulty of computing fair values, reduce agency cost as well as information asymmetry, and increase firms' ability to access the international market.

Investigating the measurement and disclosure practices based on firms' characteristics assists in determining the companies (based on tested characteristics) that need more support to overcome the problem pertaining to the low level of disclosure or undesirable measurement practices. In particular, proper measurement practice and a high level of disclosure enhance the relevance of accounting information that increase the value relevance and reduce the information asymmetry and eventually decrease agency cost.

Furthermore, the findings pointed out that the measurement practice was influenced by international activities and the intensity of BA variable, while disclosure practice was mainly influenced by international activities and audit for the Big Four of firms. This implied that the auditors who do not belong to a Big Four firm must rely on audit procedures and plans conducted by the Big Four companies. This might result in enhancing the ability of firms (that are audited by non-Big Four firms) to increase the disclosure level to provide more relevant information to stakeholders. The results of this study also provided indicators to both standards setters and responsible bodies for the purpose of inspecting the reasons behind the low level of disclosure for firms with a lower level of international activities and lower intensity of BA, in addition to their use of the cost model instead of FV to measure BA.

Future research might be needed in terms of testing the impact of other structured characteristics of companies such as legal form or listing status and sector, on mandatory and voluntary disclosure. Conducting comparative studies in the context of developing countries to examine the influence of the aforementioned determinants, owing to the availability of data, the latter investigation might only include the listed companies from several countries, in order to widen the generalizability of finding. In addition, discovering the reasons behind the low level of disclosure and using the cost model for valuing BA is deemed a worthwhile mission.

**Funding:** This research received no external funding.

**Data Availability Statement:** The data presented in this study are available on request from the corresponding author.

**Conflicts of Interest:** The author declare no conflict of interest.

## Appendix A

Questionnaire survey and disclosure index
Questionnaire survey (from financial managers)
**Company number:**

1.  Are the BA bearer plants? ☐ Yes ☐ No

2.  Is the company control the biological assets? ☐ Yes ☐ No
3.  Are the bearer plants used in the production or supply of agricultural produce?
    ☐ Yes ☐ No
4.  Do the bearer plants bear produce for more than one period? ☐ Yes ☐ No

5. Is there a remote likelihood for bearer plants of being sold as agricultural produce? ☐ Yes ☐ No

6. BA are measured based on: ☐ FV ☐ cost

**If the answer is cost, please skip question 7 to 8.**

7. If the BA are currently measured based on fair value, would these assets previously measured based on cost? ☐ Yes ☐ No

8. FV is measured based on: ☐ quoted market price in an active market (market approach)
☐ quoted prices for similar assets in active markets (market approach)
☐ quoted prices for similar assets in markets that are not active (market approach)
☐ Current replacement cost (cost approach)
☐ Present value (income approach)

9. Has the company currently or previously entitled for government grant? ☐ Yes ☐ No

10. Grant is accounted based on? ☐ IAS.20 ☐ IAS 41 ☐ Not applicable

11. Does the company translate its financial statements into foreign currency? ☐ Yes ☐ No

12. Is the company engaged in combination with other company? ☐ Yes ☐ No

13. Does the company pledge BA as security for liabilities? ☐ Yes ☐ No

14. How do you rate the level of international activities of your company?
☐ There are no international activities.
☐ Poor
☐ Good
☐ High

**Disclosure index:**
**Company number:**
Questions about companies' characteristics

1. The Intensity of biological assets:

$$= \frac{amount\ of\ biological\ assets}{amount\ of\ total\ assets}$$

2. Size: *amount of total assets*

3. Is the company financial report is audited by one of the big four audit firm? ☐ Yes ☐ No

Disclosure index

**Table A1.** Disclosure index.

| Item Number | Paragraph Number | Items | Yes | No |
|:---:|:---:|:---:|:---:|:---:|
| | | **Mandatory Items—the Entity Discloses** | | |
| 1 | 40 | An aggregate gain or loss arising during the period on initial recognition of biological assets. | | |
| 2 | 40 | An aggregate gain or loss arising during the period on initial recognition of agriculture produce. | | |
| 3 | 40 | An aggregate gain or loss arising during the period related to change in FV less costs to sell biological assets. | | |
| 4 | 42 | A narrative description of each group of biological assets. | | |
| 5 | 42 | A quantified description of each group of biological assets. | | |
| 6 | 46-a | A description of the nature of an entity's activities with each group of biological assets. | | |
| 7 | 46-b (i) | A description of non-financial measures or estimates of physical quantities of each group of the entity's BA at the end of the period. | | |
| 8 | 46-b (ii) | A description of non-financial measures or estimates of physical quantities of output of agricultural produce during the period. | | |
| 9 | 49-a | The information about BA whose title is restricted or that are pledged as security. | | |
| 10 | 49-b | The amount of commitments for the development or acquisition of biological assets. | | |
| 11 | 49-c | financial risk management strategies related to agricultural activity. | | |
| 12 | 50-a | The gain or loss arising from changes in FV less costs to sell associated with RCABA, between the beginning and the end of the period. | | |
| 13 | 50-b | Increases due to purchases associated with RCABA, between the beginning and the end of the period. | | |
| 14 | 50-c | Decreases attributable to sales and BA classified as held for sale (or included in a disposal group that is classified as held for sale) in accordance with IFRS 5, associated with RCABA, between the beginning and the end of the period. | | |
| 15 | 50-d | Decreases due to harvest associated with RCABA, between the beginning and the end of the period. | | |
| 16 | 50-e | Increases due to business combination associated with RCABA | | |
| 17 | 50-f | Net exchange differences arising on the translation of financial statements into a different presentation currency, and on the translation of a foreign operation into the presentation currency of the reporting entity | | |
| N/A | 54 | **Mandatory—additional disclosures when the FV cannot be measured reliably. The entity measures BA at cost less any accumulated depreciation and any accumulated impairment losses—the entity shall disclose.** | | |
| 18 | 54-a | A description of the biological assets. | | |
| 19 | 54-b | An explanation of why FV cannot be measured reliably. | | |
| 20 | 54-c | The range of estimates within which FV is highly likely to lie. | | |
| 21 | 54-d | The depreciation method used. | | |
| 22 | 54-e | The useful lives or the depreciation rates used. | | |
| 23 | 54-f | The gross carrying amount and the accumulated depreciation (aggregated with accumulated impairment losses) at the beginning and end of the period. | | |

**Table A1.** *Cont.*

| Item Number | Paragraph Number | Items | Yes | No |
|:---:|:---:|:---:|:---:|:---:|
| 24 | 55 | Gain or loss recognized on disposal of such biological assets | | |
| 25 | 55-a | Impairment losses, in case of disposal | | |
| 26 | 55-b | Reversals of impairment losses, in case of disposal | | |
| 27 | 55-c | The depreciation, in case of disposal | | |
| | | **Voluntary disclosures** | | |
| 28 | 43 | A quantified description of each group of BA distinguishing between consumable and bearer assets | | |
| 29 | 43 | A quantified description of each group of BA distinguishing between mature and immature assets | | |
| 30 | 51 | The amount of change in FV less costs to sell included in profit or loss due to physical changes and due to price changes | | |
| 31 | 51 | The amount of change in FV less costs to sell included in profit or loss due to physical changes and due to price changes that is presented by the group of biological assets | | |
| | | **Mandatory disclosure if the FV of BA previously measured at cost less any accumulated depreciation and impairment losses become reliably measurable during the current period—the entity discloses.** | | |
| 32 | 56-a | A description of the biological assets | | |
| 33 | 56-b | An explanation of why FV has become reliably measurable | | |
| 34 | 56-c | The effect of the change | | |
| | | **Mandatory-Government grants—the entity discloses.** | | |
| 35 | 57-a | The nature and extent of government grants recognized in the financial statements | | |
| 36 | 57-b | Unfulfilled conditions and other contingencies attaching to government grants | | |
| 37 | 57-c | Significant decreases expected in the level of government grants | | |

## Appendix B

Amendments made on previous disclosure index.

**Table A2.** Amendments made on previous disclosure index.

| Item Number | Paragraph Number of IAS 41 | Score If Item Disclosed | Items | Action Made | Note |
|---|---|---|---|---|---|
| colspan="6" | **Mandatory items—the entity discloses** | | | | |
| 1 | 40 | 1 | An aggregate gain or loss arising during the period on initial recognition of biological assets. | Amended | Minor amendment regarding language |
| 2 | 40 | 1 | An aggregate gain or loss arising during the period on initial recognition of agriculture produce. | Amended | Minor amendment regarding language |
| 3 | 40 | 1 | An aggregate gain or loss arising during the period related to change in FV less costs to sell biological assets. | Amended | Minor amendment regarding language |
| N/A | 41 | N/A | A description of each group of biological assets. | Deleted | To avoid duplication with items 4 or 5 |
| 4 | 42 | 1 | A narrative description of each group of biological assets. | Amended | Minor amendment regarding language |
| 5 | 42 | 1 | A quantified description of each group of biological assets. | Amended | Minor amendment regarding language |
| 6 | 46-a | 1 | A description of the nature of an entity's activities with each group of biological assets. | Same | |
| 7 | 46-b (i) | 1 | A description of non-financial measures or estimates of physical quantities of each group of the entity's BA at the end of the period. | Amended | Minor amendment regarding language |
| 8 | 46- b (ii) | 1 | A description of non-financial measures or estimates of physical quantities of output of agricultural produce during the period. | Amended | Minor amendment regarding language |

**Table A2.** *Cont.*

| Item Number | Paragraph Number of IAS 41 | Score If Item Disclosed | Items | Action Made | Note |
|---|---|---|---|---|---|
| N/A | 47 | N/A | The methods and assumptions applied in determining the FV of each group of agricultural produce at the point of harvest and each group of biological assets. | Deleted | As it is deleted from the content of IAS 41 |
| N/A | 48 | N/A | The FV less costs to sell agricultural produce harvested during the period, determined at the point of harvest. | Deleted | As it is deleted from the content of IAS 41 |
| 9 | 49-a | 1 | The information about BA whose title is restricted or that are pledged as security. | Same | |
| 10 | 49-b | 1 | The amount of commitments for the development or acquisition of biological assets. | Same | |
| 11 | 49-c | 1 | financial risk management strategies related to agricultural activity. | Same | |
| N/A | 50 | N/A | A reconciliation of changes in the carrying amount of BA(RCABA), between the beginning and the end of the period. | Deleted | To avoid duplication with items 11–14 |
| N/A | 50 | N/A | This reconciliation includes desegregation. | Deleted | As it not included in IAS 41 para 50 and implies many implications |
| 12 | 50-a | 1 | The gain or loss arising from changes in FV less costs to sell associated with RCABA, between the beginning and the end of the period. | Added | It is within the content of IAS 41. para 50. |
| 13 | 50-b | 1 | Increases due to purchases associated with RCABA, between the beginning and the end of the period. | Added | It is within the content of IAS 41. para 50. |

**Table A2.** *Cont.*

| Item Number | Paragraph Number of IAS 41 | Score If Item Disclosed | Items | Action Made | Note |
|---|---|---|---|---|---|
| 14 | 50-c | 1 | Decreases attributable to sales and BA classified as held for sale (or included in a disposal group that is classified as held for sale) in accordance with IFRS 5, associated with RCABA, between the beginning and the end of the period. | Added | It is within the content of IAS 41. para 50. |
| 15 | 50-d | 1 | Decreases due to harvest associated with RCABA, between the beginning and the end of the period. | Added | It is within the content of IAS 41. para 50. |
| 16 | 50-e | 1 | Increases due to business combination associated with RCABA | Added | It is within the content of IAS 41. para 50. |
| 17 | 50-f | 1 | Net exchange differences arising on the translation of financial statements into a different presentation currency, and on the translation of a foreign operation into the presentation currency of the reporting entity | Added | It is within the content of IAS 41. para 50. |
| | 54 | **Additional disclosures when the FV cannot be measured reliably. The entity measures BA at cost less any accumulated depreciation and any accumulated impairment losses—the entity shall disclose** | | | |
| 18 | 54-a | 1 | A description of the biological assets. | Same | |
| 19 | 54-b | 1 | An explanation of why FV cannot be measured reliably. | Same | |
| 20 | 54-c | 1 | The range of estimates within which FV is highly likely to lie. | Same | |
| 21 | 54-d | 1 | The depreciation method used. | Same | |
| 22 | 54-e | 1 | The useful lives or the depreciation rates used. | Same | |

**Table A2.** *Cont.*

| Item Number | Paragraph Number of IAS 41 | Score If Item Disclosed | Items | Action Made | Note |
|---|---|---|---|---|---|
| 23 | 54-f | 1 | The gross carrying amount and the accumulated depreciation (aggregated with accumulated impairment losses) at the beginning and end of the period. | Same | |
| 24 | 55 | 1 | Gain or loss recognized on disposal of such biological assets | Same | |
| 25 | 55-a | 1 | Impairment losses, in case of disposal | Same | |
| 26 | 55-b | 1 | Reversals of impairment losses, in case of disposal | Same | |
| 27 | 55-c | 1 | The depreciation, in case of disposal | Same | |
| | | | **Voluntary disclosures** | | |
| 28 | 43 | 1 | A quantified description of each group of BA distinguishing between consumable and bearer assets | Same | |
| 29 | 43 | 1 | A quantified description of each group of BA distinguishing between mature and immature assets | Same | |
| 30 | 51 | 1 | The amount of change in FV less costs to sell included in profit or loss due to physical changes and due to price changes | Same | |
| 31 | 51 | 1 | The amount of change in FV less costs to sell included in profit or loss due to physical changes and due to price changes that is presented by the group of biological assets | Same | |
| N/A | N/A | N/A | The complexity of various parameters with limited information regarding the effect on the valuation | Deleted | It is not in the content of IAS 41 |

**Table A2.** *Cont.*

| Item Number | Paragraph Number of IAS 41 | Score If Item Disclosed | Items | Action Made | Note |
|:---:|:---:|:---:|:---:|:---:|:---:|
| N/A | N/A | N/A | More information on the effects of variations in key factors | Deleted | It is not in the content of IAS 41 |
| N/A | N/A | N/A | The assumptions on future prices and costs, as well as disclosing a sensitivity analysis with multiple parameters | Deleted | It is not in the content of IAS 41 |
| **If the FV of BA previously measured at cost less any accumulated depreciation and impairment losses become reliably measurable during the current period—the entity discloses** | | | | | |
| 32 | 56-a | 1 | A description of the biological assets | Same | Only descriptive analysis |
| 33 | 56-b | 1 | An explanation of why FV has become reliably measurable | Same | Only descriptive analysis |
| 34 | 56-c | 1 | The effect of the change | Same | Only descriptive analysis |
| **Government grants—the entity discloses** | | | | | |
| 35 | 57-a | 1 | The nature and extent of government grants recognized in the financial statements | Same | Only descriptive analysis |
| 36 | 57-b | 1 | Unfulfilled conditions and other contingencies attaching to government grants | Same | Only descriptive analysis |
| 37 | 57-c | 1 | Significant decreases expected in the level of government grants | Same | Only descriptive analysis |

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
