# Peer review of "How Company Characteristics Influence Measurement Practices and Disclosure Level Prescribed within IAS 41"

_jrfm, doi:10.3390/jrfm16060288_

Round 1

Reviewer 1 Report

Thank you for the opportunity to review this paper. The paper examine an important issue relating to the IAS.41. However, it can further be improved and my suggestions as follows:

1. The introduction is very short and weak. It should be expanded by clearly highlighting the limitations of previous theoretical and empirical literature and clearly stating research contributions. Further, the introduction could be improved by explaining why this study focuses only on the general firm-level characteristics? Why, for example, corporate board characteristics [e.g., board characteristics, such as diversity (board gender, ethnicity, educational background), tenure), board size, independence, meeting, ownership structure) have not been examined in this study. It's not sufficient to examine the effect of general firm-level characteristics on the measurement practices related to assets pricing and level of disclosure required by IAS 41. Also, author should explain why Jourdan is an interesting context to be examined.

2. Literature review section is weak, and this section should be split into two parts: (i) theoretical framework, discussing the theory/ies adopted in this research and why; and (ii) empirical literature and hypothesis development.

3. The developed hypotheses should be expanded by explaining using theory/ies how each firm-level characteristic can influence study's dependent variable. Also, the developed hypotheses are general and can be applied to any context. Hence, author should link each of the developed hypotheses to the Jordanian context.

4. It's not clear what is the total sample size of the study, sample selection criteria and the final sample size.

5. The measurement of the dependent variable is not clear. How is the disclosure index developed? What items are included in the index and why? How author checked the validity and reliability of the developed index? 

6. The discussion of the obtained results is very descriptive and brief and more critical discussion is needed.

7. The conclusion is short, and author should expand the arguments by discussing the empirical, theoretical, and practical implications of their findings. Further, author is expected to expand the discussion about the weaknesses of the study and provide more avenues for future research.

Author Response

Dear Reviewer,

Please find the attached file of cover letter regarding your valuable comments. 

Kind regards

Reviewer 2 Report

The paper provides a comprehensive analysis of accounting practices in Jordanian agricultural firms and the influence of company characteristics on measurement practices related to asset pricing and level of disclosure. The research sample consists of 259 reporting companies that engage in agricultural activities, which is the entire population of such companies in Jordan. The study found that the intensity of biological assets and company size had no impact on measurement practices or level of disclosure. On the other hand, the level of international activities and audit type positively influenced measurement practices and disclosure level, with the former being more impactful in explaining measurement practices.

The paper's contribution lies in its examination of the role of company characteristics on measurement practices and level of disclosure required by IAS in the context of Jordanian agricultural firms. The results can be helpful to standards setters and regulators in developing financial reporting standards and regulations for agricultural firms in developing countries. The findings can facilitate the amending and directing of financial regulations based on determinants of measurement and disclosure practices, promoting accounting practices related to agricultural activities.

The paper also suggests that establishing an institutional structure to simplify and support the valuation of biological assets based on fair value would enhance comparability, reduce implicit costs, and increase firms' ability to access international markets. The paper suggests that future research should focus on testing the impact of other structured characteristics of companies such as legal form or listing status and sector on mandatory and voluntary disclosure. Comparative studies could also be conducted in the context of developing countries to examine the influence of these determinants.

In conclusion, the paper provides valuable insights into accounting practices in Jordanian agricultural firms and the influence of company characteristics on measurement practices and level of disclosure. The findings can be helpful in developing financial reporting standards and regulations for agricultural firms in developing countries. However, the study's generalizability may be limited to the Jordanian context, and further research may be needed to test the impact of other company characteristics and conduct comparative studies in other developing countries.

Author Response

Dear reviewer,

Please find the attached file of cover letter of your valuable comments.

I would like to thank you for your positive and valuable comments.

kind regards

Reviewer 3 Report

I would like to start by congratulating the author. The paper has a high potential for publication, but in my opinion, it should be thoroughly revised beforehand.

 1) The paper should be restructured in order to make possible that its objective becomes clear (and the interest in reading it arises) from the introduction. The abstract and the introduction should be redone so that a non-specialist can get to focus on the topic from the beginning, what exactly is going to be done in the paper?, and why it might be interesting to read it?.

 For example, from the presentation of your paper, in the introduction, and even in section 2, it seemed to me that there is a decision that companies have to make on how to value their assets (by the Fv method or by costs), is this a decision imposed or is it voluntary?, why and on the base of what factors, companies make these decisions?, how does this decision relate to the voluntary or imposed adoption of IAS 41, and how does it relate to the decision on measurement and disclosures practices?, ..... the paper does not seem to clarify these issues of mandatory measurement and disclosures and how they relate to the other decisions and features to be taken by the company until section 3. Therefore, the content of the paper should be restructured, so that all these doubts can be clear from the beginning. Please help me to see the interest of your paper from the beginning.

2) Why Jordan, what is the interest of studying the subject in Jordan? There should be an introduction to the agricultural sector in Jordan, and a description of the particularities of the sectors, what is the weight of firms in the agricultural sector?, is it dominated by small self-employed or large firms?, how affects the IAS regulation in Jordan firms? what does your case study add to the literature review?. I think this in the beginning could to help to understand the main objective of your paper to non.especialist.

3) For me, the main weakness of the paper lies precisely in the results of the regressions in section 4. I am referring to a very relevant and important issue when determining causality and biases in the estimates due to endogeneity problems.

If the entire sample used is required to submit financial reports or balance sheets in standard format and to perform different measures and disclosures, the dependent variables are already conditioned by the selection of the sample and by the type of company in question. I don't know if it is worth estimating a model to explain those variables when they are not freely determined by the decision of the companies.

I am not sure if what you are doing is forcing a simple descriptive analysis of the characteristics of the companies you have in your sample (and which have varying degrees of accountability obligations). I probably misunderstood after reading your paper. That is why I only ask you to explain it better so that other readers do not have the same doubts.

4) Continuing with the same endogeneity problem, the text shows that there are different levels of measurement and mandatory deisclosure depending on the type of companies. If this is indeed the case, there is an endogeneity problem in your regressions, for a common cause, the measurement and disclusurement requirements, insofar as these mandatory requirements are also related to, for example, the size of the firm itself or the type of firm in question. The assumptions of the general linear model are not fulfilled, and all its regressions are inconsistent, biased. You cannot use ordinary least squares. You should look for alternative estimators that avoid the inconsistency problem, you could try Instrumental variables, two-stage least squares method, or perhaps some propensity score matching technique typical of the field of program impact evaluation.

5) The hypotheses are not well specified, e.g. how are they reflected or how do they measure measurement and disclosure in the hypothesis statement?.... For example "H8 Audit type positively impacts upon the disclosure level shown by the sampled companies" is not a well specified hypothesis - audit type cannot be "positively" related to disclosure level because audit type is a categorical variable. Perhaps it refers to a special type of audit as opposed to the other types (Big Four, represented by a binary variable), but not to the categorical variable Audit type, where the categories or levels do not express more than non-measurable attributes. The same happens with the rest of the hypotheses, since they do not specify exactly what type of relationship is established between which variables. Perhaps these hypotheses should be specified after commenting on how each variable included in the hypotheses will be measured. It confused me, I only understood it after reading section 3.

Other minor issues

 6) There is an abuse of acronyms for example IFRS and IAS and BA that are used first in section 1 but are explained in section 2 (should appear and explain what it refers to from the beginning to facilitate reading for non-experts).

7) It is always interesting the analysis of the results of a survey, but you should clearly specify the source, did you do the survey your-self?, how did you do it (telephone, email)?, what was the response rate?, when was the survey conducted?, etc....

8) Please do not refer to questions that are only mentioned in the appendix, for example, what selection criteria did you use (item 1 to 5)? In this sense, Table 1 does not provide any information. Moreover, you did not mention in section 3.1 that you selected 276 companies, does this second selection criterion imply any kind of additional bias?

 Once again, I congratulate the author and encourage him to revise his current version of the paper for a better understanding.

Author Response

(The authors gave the same response as above.)

Round 2

Reviewer 1 Report

I am happy to accept the paper.

Author Response

I would like to thank you for accepting my article, and for the valuable comments and suggestions you have provided. 

Kinds regards 

Reviewer 2 Report

The paper has improved.

Reviewer 3 Report

I would like to congratulate the author for the changes introduced in this second revision that have allowed me a better understanding of the scope and relevance of his article.

 There are, however, some remaining issues that I think you have to solve before your paper can be considered for publication

 1)    It is now clear to me that you have two dependent variables. The first dependent variable is the measurement method, which is a categorical variable with two possible levels (PV and cost). I don't know if this is in fact the dependent variable used in your regression (1), because in that equation you talk about the variable "measurement practice". Actually I have not seen the definition of measurement practice anywhere. From table 5 it seem it seems that “Measuremen practice” is the same the same that “measurement methods”, a binary variable indicating 1=FV, 0 =Cost. And that may give rise to doubts, because if it is a binary or categorical variable, the estimated model (1) would NOT be the most appropriate. There should be estimated, some probability model such as probit or logistic (perhaps multinomial). I think you simply need to specify the definition of the dependent variable measurement practice (which I have not seen as specifically defined in your paper). Surely you have a definition that allows using the linear model, please explain it.

 I insist. If dependent variable is not a numeric variable, then regression 1 is not appropriate (for example you could obtain estimated probability to use FV greater than 1 and lower than 0), and I could not recommend the publication of the paper with this inconsisten linear probability regression (1).

2)  I also appreciate the explanation of the second explanatory variable disclosure level. And although I understand that it is a numerical variable (a ratio) built according to the measurement method, I do not think it is appropriate to estimate a model for the two types of companies together FV and Cost).

At least I consider that, since the construction of this dependent variable is conditioned by the measurement method, then a possible moderate effect of the measurement method should be considered. In fact, a structural change test should be carried out to find out if the coefficients estimated in regression 2 are the same for the two types of companies (FV and cost), since there is clearly a determining factor in the construction of the disclosure level variable. Using the same regression model a priori for the two types of companies is, in principle, a very restrictive hypothesis.

In short, two separate models should be estimated, one for the companies that use FV and another for the companies that use Cost, since the construction of the dependent variable is not done with the same items (the exact same thing is not being measured with this ratio), and carry out some type of contrast to find out if the estimated coefficients are identical or not (moderator effect). This would enrich the paper and test the consistency of the estimated model 2.

3) One last question is that you are working with company data, and you should test the heteroskedasticity hypothesis, and consistently estimate the standard errors of the estimated coefficients.

I hope you consider these comments as a positive review that is only intended to improve the quality of your paper.

Author Response

I would like to thank you for the valuable comments and suggestions you have provided that enriched and enhance the findings and discussion of this article.

I amended the article according to your valuable comments and suggestions

Kind regards 

Round 3

Reviewer 3 Report

Firstly, I would like to congratulate you on the effort you have put into revising your paper. I can see that you have made significant improvements to the manuscript, and it is almost ready for publication.

However, there are still a few minor changes that need to be made before the article can be accepted for publication. Specifically, there are some econometric errors that must be corrected to ensure the accuracy and validity of your findings. These errors must be addressed before the article can be considered for publication in its final version.

The specific point are the followings in your manuscript.

11)      On line 629, you begin a sentence with the word "Unfortunately." I suggest that you remove this word as it may convey an unintended negative connotation about the topic you are discussing.

 It is important to note that probabilistic models are not inherently less desirable than linear regression models. In fact, they are often more suitable for certain types of data and analyses. Therefore, the use of the word "unfortunately" in this context may not accurately reflect the scientific merit of your research and could potentially detract from the overall message of your manuscript.

22)      Another point that I would like to bring to your attention is the phrasing on line 637 of your manuscript. I strongly suggest that you remove this sentence, as it appears to be misleading and not supported by the scientific evidence.

Specifically, the sentence suggests that the assumptions related to either logistic or multiple regression tests are "ensured," which is not accurate. It is never possible to completely ensure that these assumptions are met in any statistical model. You have to test if these hypothesis are rejected. But you never (never) can say that are ensured.

Moreover, in model 2, it is clear that there is a structural change in the coefficients between FV and Cost firms, which violates one of the primary assumptions of the linear model. Therefore, I recommend that you use a more appropriate and rigorous phrasing in your manuscript to accurately reflect the limitations and challenges of your research.

33)      On line 638, you state that you implement the normal distribution in the logistic distribution function, sure?  This statement is confusing and does not seem to make sense in the context of your research. Please clarify this statement or remove it from your manuscript.

44)      on line 639, you refer to the Wald test as a test for heteroscedasticity. It is important to note that the Wald test is not used to test for heteroscedasticity, but rather for joint significance. Therefore, I recommend that you revise this statement and avoid testing heteroscedasticity in the error term. Instead, you could estimate the standard errors in a robust manner to address this issue.

55)      I suggest that you remove the paragraph from lines 637 to 640 entirely. The reason for this recommendation is that the phrasing in this section appears to reflect a lack of understanding of econometric concepts and methods.

It is always better to refrain from making inaccurate or imprecise statements that could potentially damage the scientific credibility of your research. Therefore, I strongly recommend that you eliminate this paragraph entirely from your manuscript.

66)      on line 660, you present a formulation that does not represent a logistic regression model but rather a linear regression model. It is crucial that you review the standard formulation for logistic regression in any econometrics textbook to ensure that your model specification is accurate.

In its current form, your model (1) does not align with the estimation procedure presented in your manuscript, which could negatively impact the scientific integrity of your research. Therefore, I strongly recommend that you revise this section to accurately reflect the model you are estimating.

77)      I would like to acknowledge your efforts in defining the variable "Measurement practice" on line 554 of your manuscript. I appreciate your attention to detail in clarifying this variable's meaning and its role in the estimation of the probability of Measurement Practice being FV.

 However, I would like to recommend that you revisit Table 8 and modify the title of the dependent variable to reflect the probability of Measurement Practice being FV explicitly. This change will enhance the clarity of your results and help readers better understand the dependent variable's nature.

88)      An interesting and valuable results of your logistic regression analysis: internationalization and business activity intensity are the primary drivers of firms' use of fair value (FV) measurement. This information is particularly insightful and help to  understand the determinants of accounting measurement practices.

99)      I have noticed an issue with the naming of the "Audit type" variable used in your analysis. As a categorical variable, it is difficult to interpret the coefficients when referring to "Audit type." Therefore, I suggest that you rename this variable to "Audit for Big Four." In fact, it is not the effect of the “Audit type” but  the impact of been audit by a Big Four on overall disclosure that is quite substantial, with firms audited by one of the Big Four firms showing a 12.5% increase in overall disclosure (as shown in Table 10). This variable has the largest effect size on overall disclosure, making it a critical variable in your analysis.

I recommend making this change in the name of the variable to improve the clarity and interpretation of your results.

110)   It is commendable to have run the two models separately. However, the results of model 2, model 2.1, and model 2.2 should be presented in the same table (in different columns)

111)   Additionally,  the Chow test for structural change should be applied (only the SSR of model 2, model 2.1, and model 2.2 is needed) to verify whether the models are indeed different and whether the behavior differs between FV and Cost companies. Please consult any econometrics manual on how to perform this test as its result is very significant and has important implications for the study. If the Chow test indicates a difference between model 2.1 and model 2.2, then model 2 is inconsistent (among other things because one of the assumptions of multiple linear regression on fixed coefficients throughout the sample is violated), and no conclusions can be drawn based on it (table 10). Instead, separate conclusions should be drawn for the two types of companies.

112)   As I mentioned before, there is still a potential issue with heteroscedasticity in Model 2. Therefore, the coefficients of Models 2, 2.1, and 2.2 should be estimated robustly. In R, the sandwich library should be used, while in Stata, setting se="ro" is sufficient.

I hope you find this feedback helpful. Overall, I believe that your research is promising, and I look forward to seeing the final version of your article once the necessary corrections have been made.

Author Response

I would like like to thank you for your valuable comments.

I have made all necessary corrections according to your valuable suggestions and comments. 

Kind regards

Round 4

Reviewer 3 Report

I can only congratulate the author.

My comments have always been aimed towards improving the paper, and I am grateful for the good spirit with which the author has worked. I think the author will have noticed that I have taken the revision of the paper very seriously, and I have dedicated the time and work it deserved. In my opinion, the paper was of great interest and had a high potential, especially because of the field work that the author had done. However, I felt obliged to point out the errors that I have detected in the successive versions of the paper.

I believe that the paper, in my opinion, has now in this last version a high quality and scientific rigor, and therefore it is a pleasure for me to recommend its publication in its current state.